

# Knowledge, attitudes, and practices (KAPs) regarding tick-borne rickettsial disease among medical staff in endemic areas of China

Miaohui Shao[1,2,*], Peiyuan Hu[1,3,*], Pengpeng Xu[2], Jie Sun[2], Xiaqing Zhang[1,3], Dan Zhang[4], Yong Shen[4], Dawei Gao[2], Weigang Zhang[2], Wei Qin[2] and Yong Lyu[1,2]

[1] School of Public Health, Anhui Medical University, Hefei, Anhui, China
[2] Lu'an Municipal Center for Disease Control and Prevention, Lu'an, Anhui, China
[3] Anhui Provincial Key Laboratory of Microbiology & Parasitology, Hefei, Anhui, China
[4] Lu'an Hospital of Anhui Medical University, Lu'an, Anhui, China
* These authors contributed equally to this work.

Corresponding author
Yong Lyu, lyong@lacdc.com.cn

## ABSTRACT

Tick-borne rickettsial disease (TBRD) is a perilous acute infection that often eludes diagnosis in its early stages. The triad of knowledge, attitudes, and practices (KAPs) among medical professionals is key to reducing missed diagnosis rates. Therefore, a meticulous evaluation of KAPs is imperative. This study aimed to delve into the understanding of TBRD and explore the beliefs and practices related to personal prevention methods among individuals in Lu'an, a hotspot for TBRD. During the summer months of 2023, convenience sampling was employed by circulating a confidential questionnaire to 1,206 participants in the endemic regions of China. This questionnaire painted a comprehensive picture of the participants' sociodemographic profiles and their KAPs levels vis-à-vis TBRD. The findings revealed that participants scored a mere 55.78% in knowledge, while their attitudes and practices garnered impressive scores of 90.09% and 90.83%, respectively. Upon further analysis using multiple linear regression, several intriguing patterns emerged. Male participants, employed in the Infectious Disease Department, held vice-senior or higher titles, or had prior medical training demonstrated superior knowledge scores. On the other hand, medical personnel who were younger than 30, possessed graduate degrees or higher qualifications, and had training excelled in attitudes and practices. Notably, when employing the Boston Consulting Group (BCG) matrix, a significant distribution of medical personnel was observed across the four quadrants. Specifically, 37.43%, 13.19%, 19.61%, and 29.77% fell into the first, second, third, and fourth quadrants. This survey underscores the commendable attitudes and practices of medical staff towards TBRD in endemic regions of China. However, their knowledge level remains wanting and demands urgent improvement.

## INTRODUCTION

Tick-borne rickettsial disease (TBRD) refers to rickettsial diseases caused by tick transmission. Rickettsial is divided into the Rickettsiaceae and Anaplasma families, and Rickettsiaceae is further divided into four categories, namely, the spotted fever group rickettsial (SFGR), typhus group rickettsial, ancestral group rickettsial, and transitional group rickettsial (*Merhej et al., 2014*). The disease is transmitted *via* saliva through a tick bite (*Nicholson et al., 2019*; *Walker & Ismail, 2008*). More than 20 causative agents have been identified (*Kernif et al., 2016*). Infections are often similar to many other acute febrile illnesses characterized by fever, chills, and headache, with maculopapular rash at the site of the vector bites similar to many other infections (*Adem, 2019*; *Biggs et al., 2016*). Some patients may present with erythema nodosa in the early stage (*Peregrina-Rivas et al., 2022*), which can be used as a reference for early identification. In a small number of individuals, there have been reported cases of concurrent infections with two types of SFGR, occasionally accompanied by severe sequelae such as encephalitis (*Mendes et al., 2021*; *Rudakov et al., 2019*).

In recent years, studies have revealed that missed diagnosis and misdiagnosis of TBRD are prevalent and have a significant impact on the prognosis of patients (*Biggs et al., 2016*; *Igolkina et al., 2022*). One of the most important factors in the misdiagnosis of TBRD is the lack of awareness among primary care staff and emergency department physicians when encountering undifferentiated acute febrile disease during the tick season (*Walker et al., 2022*). According to the investigation, many patients do not exhibit rash during diagnosis and treatment (*Igolkina et al., 2022*; *Kinoshita et al., 2021*). A rash's delay in appearance or absence can also make a TBRD diagnosis challenging. Literature reviews have shown that many cases of delayed disease diagnosis were caused by a lack of knowledge among medical staff (*Bégué, 2023*; *Li & Liu, 2021*). Early recognition in the clinical course is critical because this is the most effective period of antibacterial therapy (*Biggs et al., 2016*). Medical staff play an important role in detecting and controlling infectious diseases. They are also an important source of information to help raise public awareness among high-risk people (*Eleftheriou et al., 2023*). The Knowledge, Attitudes, and Practices (KAPs) theory is the most widely utilized framework for explaining how individual knowledge and beliefs influence change in health practices. Therefore, gaining a comprehensive understanding of the KAPs of medical personnel towards TBRD is imperative.

China is a country with a vast territory and lush forests. With the intensification of globalization, climate change, and expansion of human activities, the incidence of TBRD in China has gradually increased, and the incidence range has continued to expand (*Fang et al., 2015*). Lu'an City in Anhui Province is an endemic area in China. Lu'an City is located in the Ta-pieh Mountain area. This city is rich in forest vegetation and suitable for tick breeding. As early as 2006, tick-borne diseases appeared in Anhui Province (*Zhang, Ni & Feng, 2010*), and in 2013, it was confirmed that SFGR was transmitted in Lu'an City (*Hu et al., 2022*; *Lyu et al., 2021*). Follow-up investigations by the local Centers for Disease Control and Prevention showed that there was a natural foci of TBRD in Lu'an City,

ehrlichiosis, as the TBRD also has been found to exist in Lu'an and hundreds of cases of TBRD are diagnosed annually in Lu'an (*Lu et al., 2023*).

Increasing awareness among professionals in the field is paramount for establishing effective control measures against pathogens. Assessing the knowledge of medical staff could provide useful information on the levels of interventions required to adapt government-related measures to meet the demands of a changing world. Nevertheless, to our knowledge, there are no KAP data for specific occupational high-risk populations in China. The survey on KAPs in China mainly covers the general population in endemic areas (*Lyu et al., 2017*). Therefore, this study aimed to assess the basic knowledge of medical staff in endemic areas of China. This approach will be essential for building core competencies regarding TBRD and suggesting additional adjustment measures.

# MATERIALS AND METHODS

## Ethics committee
The Health Commission of Lu'an City (Health and Disease Control Secretary (2023) No. 20) approved this study for implementation. The study was conducted in full compliance with Chinese legal and regulatory requirements.

## Study area
A quantitative descriptive cross-sectional study was conducted among medical staff in Lu'an. Survey locations were selected based on high annual TBRD incidence rates (*Lu et al., 2023*). According to the principle of convenience sampling, 21 medical institutions in Lu'an were selected to be investigated.

## Study population
We selected medical personnel from various medical institutions, including doctors, nurses, inspectors, and other relevant professionals. All eligible medical personnel were welcome to participate, except those unable to use a mobile phone, as our questionnaire required a smartphone for completion. Furthermore, participants were required to be at least 18 years of age.

## Sample size
The data on TBRD were collected through a structured questionnaire from July to August 2023. The questionnaire content was carefully formulated by experts following thorough discussions and research. The sample size for the study area was determined using the formula for cross-sectional studies (Eq. (1)).

$$n = \frac{z_\alpha^2 \times pq}{d^2}; n = \frac{(1.96)^2 \times 0.3(1-0.3)}{(0.03)^2}; n = 896 \approx 900 \tag{1}$$

where $n$ = sample size, $Z$ = 1.96 (95% confidence level), $P$ = expected prevalence or proportion (in proportion of one; 30%, $P = 0.3$), and $d$ = precision (in proportion of 1; whereas $P = 0.3$, therefore $d = 0.03$).

Considering the possibility of no response or questionnaire failure in this study, the actual sample size was increased by 1.5 times and determined to be 1,350. Multiple random sampling techniques were applied to select the study participants. The only unwillingness to contribute was the exclusion criteria.

### Study instrument

A preliminary survey based on a literature review and expert suggestions was adopted before the investigation. The final questionnaire was determined after several rounds of modification in strict accordance with the requirements of the questionnaire design. To ensure the reliability of the questionnaire results, this study adopted an on-site survey completed by professionals with on-site guidance, and the questionnaire data were collected uniformly in the background.

## METRICS

### Questionnaire content

The structured questionnaire was comprised of four sections. First, the socio-demographic section included the variables of medical staff's gender, age, department, education level, professional title, hospital level, occupation, and training on TBRD. Second, the knowledge section of the questionnaire assessed medical staff on their knowledge of TBRD, including symptoms of TBRD, incubation period, susceptible subjects, duration of treatment, preferred drug, and high incidence season. This section included nine close-ended questions, each with several possible answers. One point was given for a correct answer, and no points were given for a wrong answer. The possible score range for the knowledge section was zero to nine points. Third, the attitudes section contained questions about medical staff's impressions towards TBRD. An attitude was defined as "a complex mental state involving beliefs, feelings, values, and dispositions to act in certain ways" (*Altmann, 2008*). This section included seven questions that each had five possible answers. A five-point rating scale (*Singh et al., 2023*) (Strongly agree = 5; Moderately agree = 4; Neutral = 3; Partially disagree = 2; Strongly Disagree = 1) was used to evaluate the medical staff's attitudes. The possible score range for the attitudes section was zero to 35 points. Finally, section four comprised seven questions regarding TBRD practices by medical staff. The scoring method was the same as above. The possible score range for the practices section was zero to 35 points. Additionally, Cronbach's coefficient of this questionnaire, which evaluated the aspects of attitudes and practices, was 0.898.

### Statistical analysis

SPSS® and Microsoft Excel® software were used for statistical analysis. We applied multivariate analysis, one-way variance analysis (ANOVA), and Student's t test. The Boston Consulting Group (BCG) determined the key health education population.

## RESULTS

### Demographic data

A total of 1,350 questionnaires were distributed, and 1,206 valid questionnaires were recovered, for an effective response rate of 89.26%. Women accounted for 59.12% of the respondents. Most of the participants were under 30 years old (34.00%). In terms of lower and higher levels, educational background was almost always higher among the respondents (undergraduate course or above, 66.42%). Most respondents were either working in tertiary hospitals (40.22%) or secondary hospitals (29.60%) at the respective study locations. Approximately half of the respondents (41.63%) were clinicians, 5.56% were inspection personnel, and 37.81% were nurses. More people had participated in general training on tick-borne diseases than those who had not, accounting for 73.05%. For details, please see Table 1.

### Scoring of medical staff

The total average score of the 1,206 medical staff was 68.25 ± 7.31, for a total score of 86.39%. The average score of the knowledge dimension was 4.93 ± 1.22, and the scoring rate was 55.78%. However, the average score of the attitudes dimension was 31.53 ± 3.70 for a scoring rate of 90.09%, and the average score of the practices dimension was 31.79 ± 3.64 for a scoring rate of 90.83%.

### One-way analysis of KAPs scores of TBRD among medical staff

Table 2 shows the results of the single-factor analysis. The results showed that sex, age, professional title, department, and training influenced the TBRD knowledge score. Age, education level, working years, and training influenced the TBRD attitudes score. Age, education level, working years, position, and training influenced the TBRD practices score. The differences were statistically significant ($P < 0.05$).

### Multivariate analysis of KAPs scores of TBRD among medical staff

Multiple linear regression was carried out with the medical staff members' knowledge, attitudes, and practices related to TBRD as the dependent variables and the meaningful variables identified in the univariate analysis as the independent variables. The variable assignments are shown in Table 3. Each variable's variance inflation factor (VIF) was less than 10, and there was no multicollinearity.

The results showed that sex, department, title, and training of medical staff influenced knowledge scores ($P < 0.05$), specifically reflected in the higher scores for males, infectious disease department personnel, people with a vice-senior or higher titles, and trained medical staff.

Notably, the magnitude of the absolute value of the regression coefficient directly correlates with the extent of influence the independent variable exerts on the dependent variable. A larger absolute value signifies a greater impact. Therefore, gender has the most significant influence on knowledge scores, whereas department exhibits the least impact. Age, education level, and training of medical personnel influenced the attitudes and practices scores ($P < 0.05$), specifically reflected in the higher scores of medical personnel

**Table 1 Socio-demographic and other characteristics of study participants ($n$ = 1,206).**

| Characteristic | Number (%) | Characteristic | Number (%) |
|---|---|---|---|
| Gender | | Hospital type | |
| Male | 493 (40.88) | Village clinics/Township health centers | 270 (22.39) |
| Female | 713 (59.12) | Centers for disease control and prevention | 94 (7.79) |
| Age group (year) | | Secondary hospital | 357 (29.60) |
| <30 | 410 (34.00) | Tertiary hospital | 485 (40.22) |
| 30–40 | 405 (33.58) | department | |
| >40 | 391 (32.42) | Infectious diseases department | 73 (6.05) |
| Level of education | | Public health department | 67 (5.56) |
| Technical secondary school and below | 120 (9.95) | Laboratory department | 65 (5.39) |
| Junior college | 285 (23.63) | Outpatient and emergency department | 206 (17.08) |
| Undergraduate | 742 (61.53) | Internal medicine | 361 (29.93) |
| Postgraduate or above | 59 (4.89) | Surgery | 206 (17.08) |
| Professional title | | Other | 228 (18.91) |
| No professional title | 195 (16.17) | Occupation | |
| Primary title | 469 (38.89) | Clinician | 502 (41.63) |
| Intermediate title | 379 (31.43) | Nurse | 456 (37.81) |
| Vice-senior title or high title | 163 (13.51) | Inspection personnel | 67 (5.56) |
| Years of experience | | Public health doctor | 76 (6.30) |
| 10 | 569 (47.18) | Other | 105 (8.70) |
| 10–20 | 325 (26.95) | Training provided | |
| >20 | 312 (25.87) | Yes | 881 (73.05) |
| | | No | 325 (26.95) |

**Table 2 Results of single factor analysis of KAP scores of TBRD among medical staff.**

| Variable | | Knowledge | | | Attitude | | | Practices | | |
|---|---|---|---|---|---|---|---|---|---|---|
| | | Score | t/F | P | Score | t/F | P | Score | t/F | P |
| Gender | | | 2.916 | 0.004 | | −0.484 | 0.629 | | −0.953 | 0.341 |
| | Male | 5.05 ± 1.21 | | | 31.46 ± 3.56 | | | 31.67 ± 3.51 | | |
| | Female | 4.85 ± 1.22 | | | 31.57 ± 3.80 | | | 31.88 ± 3.73 | | |
| Age group (year) | | | 2.003 | 0.043 | | 3.169 | 0.000 | | 3.530 | 0.000 |
| | <30 | 4.89 ± 1.23 | | | 32.12 ± 3.84 | | | 32.27 ± 3.84 | | |
| | 30–40 | 4.95 ± 1.27 | | | 31.54 ± 3.68 | | | 31.96 ± 3.54 | | |
| | >40 | 4.96 ± 1.16 | | | 30.89 ± 3.48 | | | 31.13 ± 3.42 | | |
| Level of education | Level of education | | 1.119 | 0.347 | | 2.504 | 0.000 | | 3.298 | 0.000 |
| | Technical secondary school and below | 4.83 ± 1.17 | | | 30.42 ± 3.75 | | | 30.43 ± 3.74 | | |
| | Junior college | 4.87 ± 1.19 | | | 31.29 ± 3.71 | | | 31.61 ± 3.56 | | |
| | Undergraduate | 4.96 ± 1.21 | | | 31.77 ± 3.68 | | | 32.05 ± 3.65 | | |
| | Graduate or above | 5.07 ± 1.60 | | | 31.83 ± 3.41 | | | 32.27 ± 3.01 | | |

| Variable | | Knowledge | | | Attitude | | | Practices | | |
|---|---|---|---|---|---|---|---|---|---|---|
| | | Score | t/F | P | Score | t/F | P | Score | t/F | P |
| Hospital type | | | 0.766 | 0.633 | | 1.175 | 0.264 | | 1.112 | 0.338 |
| | Village clinics/township health centers | 4.86 ± 1.20 | | | 31.24 ± 3.79 | | | 31.27 ± 3.83 | | |
| | Centers for disease control and prevention | 5.24 ± 1.33 | | | 31.04 ± 3.53 | | | 31.48 ± 3.60 | | |
| | Secondary hospital | 4.85 ± 1.16 | | | 32.00 ± 3.67 | | | 32.33 ± 3.56 | | |
| | Tertiary hospital | 4.97 ± 1.25 | | | 31.43 ± 3.69 | | | 31.75 ± 3.55 | | |
| Years of experience | | | 1.560 | 0.132 | | 2.517 | 0.000 | | 2.819 | 0.000 |
| | <10 | 4.90 ± 1.25 | | | 31.98 ± 3.74 | | | 32.25 ± 3.68 | | |
| | 10–20 | 4.98 ± 1.22 | | | 31.23 ± 3.83 | | | 31.51 ± 3.67 | | |
| | >20 | 4.95 ± 1.16 | | | 31.01 ± 3.40 | | | 31.25 ± 3.42 | | |
| Professional title | | | 2.831 | 0.004 | | 1.306 | 0.160 | | 1.587 | 0.065 |
| | No professional title | 4.72 ± 1.15 | | | 31.43 ± 3.35 | | | 31.57 ± 3.49 | | |
| | Primary title | 4.92 ± 1.21 | | | 31.76 ± 4.14 | | | 31.96 ± 4.03 | | |
| | Intermediate title | 4.94 ± 1.22 | | | 31.51 ± 3.39 | | | 31.83 ± 3.30 | | |
| | Vice-senior title or high title | 5.20 ± 1.28 | | | 31.02 ± 3.43 | | | 31.52 ± 3.35 | | |
| Occupation | | | 1.558 | 0.133 | | 1.260 | 0.193 | | 1.829 | 0.000 |
| | Clinician | 4.85 ± 1.20 | | | 31.66 ± 3.75 | | | 31.94 ± 3.68 | | |
| | Nurse | 4.93 ± 1.24 | | | 31.50 ± 3.71 | | | 31.79 ± 3.44 | | |
| | Inspection personnel | 4.99 ± 1.14 | | | 31.91 ± 3.15 | | | 32.21 ± 3.30 | | |
| | Public health doctor | 5.24 ± 1.23 | | | 31.48 ± 3.31 | | | 31.71 ± 3.46 | | |
| | Other | 5.10 ± 1.25 | | | 30.78 ± 4.00 | | | 30.91 ± 4.42 | | |
| Department | | | 5.368 | 0.000 | | 1.338 | 0.140 | | 1.576 | 0.068 |
| | Infectious diseases department | 5.64 ± 1.35 | | | 31.25 ± 3.55 | | | 32.11 ± 2.92 | | |
| | Public health department | 5.49 ± 1.15 | | | 32.16 ± 2.99 | | | 32.28 ± 3.12 | | |
| | Laboratory department | 5.02 ± 1.18 | | | 31.38 ± 4.49 | | | 31.66 ± 4.72 | | |
| | Outpatient and emergency department | 4.71 ± 1.09 | | | 31.27 ± 3.76 | | | 31.65 ± 3.50 | | |
| | Internal medicine | 4.90 ± 1.15 | | | 31.85 ± 3.46 | | | 32.02 ± 3.63 | | |
| | Surgery | 4.93 ± 1.24 | | | 31.68 ± 3.58 | | | 31.81 ± 3.54 | | |
| | Other | 4.76 ± 1.29 | | | 31.04 ± 4.05 | | | 31.35 ± 3.85 | | |
| Training provided | | | 3.523 | 0.000 | | 3.171 | 0.002 | | 2.897 | 0.004 |
| | Yes | 5.01 ± 1.19 | | | 31.73 ± 3.63 | | | 31.98 ± 3.61 | | |
| | No | 4.73 ± 1.27 | | | 30.97 ± 3.85 | | | 31.30 ± 3.63 | | |

younger than 30 years old, participants with postgraduate or higher education, and trained medical personnel. Amongst the various influences, training has the most profound effect on attitudes and practices, whereas education's impact on these aspects is comparatively minimal (Table 4).

**Table 3 Variable assignment table.**

| Variable | Assignment |
|---|---|
| Gender | Male = 1; Female = 2 |
| Age group (year) | <30 = 1; 30–40 = 2; >40 = 3 |
| Department | Infectious diseases department = 1; Public health department = 2; Laboratory department = 3; Outpatient and emergency department = 4; Internal medicine = 5; Surgery = 6; Other = 7 |
| Professional title | No professional title = 1; Primary = 2; Intermediate = 3; Vice-senior title or high title = 4 |
| Level of education | Technical secondary school and below = 1; Junior college = 2; Undergraduate = 3; Postgraduate or above = 4 |
| Years of experience | <10 = 1; 10–20 = 2; >20 = 3 |
| Occupation | Clinician = 1; Nurse = 2; Inspection personnel = 3; Public health doctor = 4; Other = 5 |
| Training provided | Yes = 1; No = 2 |

**Table 4 Results of multiple linear regression of KAP scores of TBRD among medical staff.**

| Variable | | Regression coefficient | Standard error | t | P | Standardized regression coefficient |
|---|---|---|---|---|---|---|
| Knowledge | Constant | 5.946 | 0.232 | 25.605 | 0.000 | |
| | Gender | −0.234 | 0.073 | −3.211 | 0.001 | −0.094 |
| | Age | −0.086 | 0.049 | −1.730 | 0.084 | −0.057 |
| | Department | −0.101 | 0.021 | −4.852 | 0.000 | −0.138 |
| | Professional title | 0.127 | 0.043 | 2.966 | 0.003 | 0.096 |
| | Training provided | −0.228 | 0.079 | −2.871 | 0.004 | −0.083 |
| Attitude | Constant | 32.897 | 0.654 | 50.285 | 0.000 | |
| | Age | −0.766 | 0.255 | −3.010 | 0.003 | −0.169 |
| | Level of education | 0.396 | 0.150 | 2.634 | 0.009 | 0.078 |
| | Years of experience | 0.163 | 0.255 | 0.638 | 0.524 | 0.036 |
| | Training provided | −0.928 | 0.240 | −3.860 | 0.000 | −0.111 |
| Practice | Constant | 32.943 | 0.661 | 49.817 | 0.000 | |
| | Age | −0.531 | 0.250 | −2.120 | 0.034 | −0.119 |
| | Level of education | 0.481 | 0.148 | 3.252 | 0.001 | 0.097 |
| | Years of experience | −0.009 | 0.251 | −0.036 | 0.972 | −0.002 |
| | Occupation | −0.148 | 0.084 | −1.765 | 0.078 | −0.050 |
| | Training provided | −0.817 | 0.236 | −3.459 | 0.001 | −0.100 |

## Analysis of the TBRD health education population

To further characterize the TBRD health education of medical workers, knowledge of TBRD was shown as the horizontal axis, and practice was shown as the vertical axis. Four-quadrant analysis was used. A total score of nine points was considered to indicate knowledge, and scores ≥5 were considered to indicate quality. The median attitudes score was 33, and a positive attitude was defined as a score ≥33. The median practices score was 34, and ≥34 was considered correct practices. The analysis revealed that 37.43%, 13.19%,

**Table 5 Relationships among knowledge, attitude, and practices related to tick-borne rickettsial disease.**

| Quadrant | Knowledge+practice | Attitude of TBRD | | Total |
|---|---|---|---|---|
| | | Active (%) | Negative (%) | |
| I | Correct+correct | 377 (88.71) | 48 (11.29) | 425 (35.24) |
| II | Error+correct | 165 (89.19) | 20 (10.81) | 185 (15.34) |
| III | Error+error | 21 (9.50) | 200 (90.50) | 221 (18.33) |
| IV | Correct+error | 41 (10.93) | 334 (89.07) | 375 (31.09) |
| | Total | 604 (50.08) | 602 (49.92) | 561 (100.00) |

19.61%, and 29.77% of the medical staff fell into the first, second, and fourth quadrants, respectively. The population's attitude accuracy rate in each quadrant is detailed in Table 5.

## DISCUSSION

TBRD is a regional disease whose geographical distribution is synchronized with the seasonal activities of ticks as vectors (*Romer et al., 2020*). In recent years, TBRD and the discovery of new pathogens have been reported in more and more countries, making it a public health issue of worldwide concern (*Faccini-Martinez et al., 2018*; *Leon de la Fuente & Sanchez, 2018*; *Sevestre et al., 2021*; *Silva-Ramos et al., 2022*). Improving the medical staff's understanding of TBRD is conducive to improving the disease's early diagnosis rate and enhancing patients' recovery levels (*Walker et al., 2022*). This study provides new insights into TBRD awareness among healthcare workers in endemic areas of China, as well as new practices for preventing tick bites and controlling tick transmission.

The results showed that the medical staff was poor in awareness of TBRD. This could be attributed to the scarcity of health education initiatives. As a newly emerging infectious disease, TBRD has not yet been fully managed in China (*Lyu et al., 2021*). Compared to the results of this survey, other studies have found that the knowledge, attitudes, and practices of medical staff were significantly different. In a survey on tick-borne diseases in endemic areas of the United States, the attitudes and practices levels of the local population were low, reflected in the low participation rate in personal prevention practices (*Beck et al., 2022*). A study in Finland showed that tick prevention practices in the region is still in its infancy (*Zöldi et al., 2017*). In contrast, many Czech students used targeted preventive measures regarding tick-borne diseases before going out (*Butler et al., 2016*).

Further analysis of our results showed that gender, age, professional title, department, and training affected the TBRD knowledge score, mainly reflected in the higher scores of male participants, infectious department personnel, having postgraduate or higher education, and trained medical staff. The infectious department is a common department for TBRD patients, so relevant medical staff are expected to know more about the disease. At the same time, as the job titles of medical staff increase, so do their scores. Senior medical staff in China have greater professional knowledge and rich experience, so their scores are higher than those of lower-ranking medical staff. It can be seen from a survey of students in Egypt (*Khamassi Khbou et al., 2020*) that after training in a specific course,
students gained a higher level of knowledge, which is consistent with our findings. Training on TBRD in endemic areas of China is increasing, which is a good sign. Notably, men's TBRD knowledge scores were higher than women's. The reasons behind this are worth further reflection.

Some unexpected results are also worth highlighting. First, when analyzing the attitudes and practices of medical workers with rickettsial disease, we found that age, education level, and training were the most common factors influencing attitudes and practices scores. As age decreased and education increased, the score of medical personnel was higher. As education level increased, knowledge level increased. Compared with the results of a study on tick-borne diseases and German foresters (*San Martin Rodriguez et al., 2020*), the education level of medical personnel in endemic areas in China did not affect the knowledge score but played an important role in the attitudes and practices scores. In Germany, education did not affect the practices of those surveyed. However, surveys in prevalent areas of the United States showed that education was associated with better practices (*Beck et al., 2022*). In addition, the younger age group showed better attitudes and practices than the older age group, suggesting that more attention should be paid to the practices of the older group in daily work. Finally, the notably higher scores obtained by trained medical staff further highlighted the importance of relevant training.

When analyzing TBRD health education groups, we found that 35.24% of the people fell into the first quadrant, and 88.71% of the first quadrant had a positive attitude toward TBRD. This part of the population is a successful product of TBRD health education (*Brockmann & Brezinski, 2015*; *Ercis & Unalan, 2015*). TBRD education must continue to be highlighted in future efforts to educate medical professionals. A total of 15.34% of the population fell into the second quadrant, with poor knowledge but superior practice levels. This part of the population under-performed in TBRD health education, and in-depth analysis is needed to analyze the causes. A total of 18.33% of the population fell into the third quadrant, with poor knowledge and practices. A total of 31.09% of the population fell into the fourth quadrant, indicating that while they showed good level of knowledge, they had no corresponding practice ability, suggesting that their knowledge did not produce an effect.

There were some limitations to this study. We selected Lu'an City in Anhui Province as an endemic area in China to conduct a cross-sectional survey. The survey site needs to be further expanded to be more representative of the population. At the same time, this study was conducted as a field survey, and the options about attitudes and practices were graded, which could easily lead to the medical staff choosing inconsistent options to obtain higher scores.

## CONCLUSION

Overall, this KAPs study provides information on the knowledge and practice of TBRD among healthcare workers in endemic areas of China and analyzes its influences. Given the low knowledge scores of medical personnel, further intervention and education programs are needed to reduce the misdiagnosis rate of TBRD further. Further epidemiological studies also need to be performed increase the number of medical professionals in the first

quadrant while also strengthening the health intervention of the population in the third quadrant to promote the virtuous circle of knowledge, belief, and practice of medical staff.

## ACKNOWLEDGEMENTS

The authors thank all clinicians who contributed data from Jinzhai County People's Hospital, Huoshan County People's Hospital, Huoqiu County People's Hospital, Shucheng County People's Hospital, and Lu'an City People's Hospital.

### Funding

This study was supported by the Scientific Research Projects of Health Commission of Anhui Province in 2022 (AHWJ2022b119) and the Anhui Provincial Key Laboratory of Pathogenic Biology (BY. 2022Z05). The funders had no role in study design, data collection and analysis, decision to publish, or preparation of the manuscript.

### Grant Disclosures

The following grant information was disclosed by the authors:
Health Commission of Anhui Province: AHWJ2022b119.
Anhui Provincial Key Laboratory of Pathogenic Biology: BY. 2022Z05.

### Competing Interests

The authors declare that they have no competing interests.

### Author Contributions

- Miaohui Shao conceived and designed the experiments, performed the experiments, analyzed the data, prepared figures and/or tables, authored or reviewed drafts of the article, and approved the final draft.
- Peiyuan Hu analyzed the data, authored or reviewed drafts of the article, and approved the final draft.
- Pengpeng Xu conceived and designed the experiments, prepared figures and/or tables, and approved the final draft.
- Jie Sun analyzed the data, prepared figures and/or tables, and approved the final draft.
- Xiaqing Zhang conceived and designed the experiments, prepared figures and/or tables, and approved the final draft.
- Dan Zhang performed the experiments, authored or reviewed drafts of the article, and approved the final draft.
- Yong Shen conceived and designed the experiments, performed the experiments, authored or reviewed drafts of the article, and approved the final draft.
- Dawei Gao performed the experiments, authored or reviewed drafts of the article, and approved the final draft.
- Weigang Zhang conceived and designed the experiments, prepared figures and/or tables, and approved the final draft.

Peer J

- Wei Qin analyzed the data, authored or reviewed drafts of the article, and approved the final draft.
- Yong Lyu conceived and designed the experiments, analyzed the data, prepared figures and/or tables, authored or reviewed drafts of the article, and approved the final draft.

### Human Ethics

The following information was supplied relating to ethical approvals (*i.e.*, approving body and any reference numbers):

The research protocols used in this study were reviewed and approved by the Health Commission of Lu'an City (Health and Disease Control Secretary [2023] No. 20) and written informed consent was obtained from all research participants.

### Data Availability

The raw measurements are available in the Supplemental File.

### Supplemental Information

Supplemental information for this article can be found online at http://dx.doi.org/10.7717/peerj.17562#supplemental-information.

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
