# Peer review of "Knowledge, attitudes, and practices (KAPs) regarding tick-borne rickettsial disease among medical staff in endemic areas of China"

_PeerJ, doi:10.7717/peerj.17562_

## Round 0.1 · original submission · Major Revisions

Please submit a revised version of the manuscript along with a point-by-point rebuttal letter addressing the reviewers' comments in detail.

**Language Note:** The review process has identified that the English language must be improved. PeerJ can provide language editing services - please contact us at [email protected] for pricing (be sure to provide your manuscript number and title). Alternatively, you should make your own arrangements to improve the language quality and provide details in your response letter. – PeerJ Staff

Reviewer 1 ·

Basic reporting

A) Unclear, ambiguous, with no professional English language used throughout the manuscript
B) Introduction & background need to be improved:

I recommend to improve the description in lines 57- 86 to provide more justification for your study (specifically, you should expand upon the knowledge gap being filled).

- Reorganization of references for improving clarity about scientific background is necessary. For instance, in line 44 Bánovic et al., (2023) are referenced, but they used as source the Merhej et al., (2014) study. I highly recommend expanding the theoretical background of this section by including seminal references (see Parola P et al. 2005. Doi:10.1051/vetres:2005004)
- Line 45. Specify the tick genera. Needs better references addressing human infections (i.e. check for Walker D & Ismail N., 2008 Nature Rev Microbiol. doi:10.1038/nrmicro1866; Nicholson WL et al., 2019. Doi:10.1016/B978-0-12-814043-7.00027-3), Nasirian et al. 2023 the reference you used targeted infections in domestic ruminants.
- Line 45. Correct grammar. “Pathogenic pathogens”?
- Line 47. Headache instead of headaches. The inoculation eschar is not often found it in rickettsial diseases such as the produced by R. rickettsii, R. massiliae or R. parkeri
- Line 48. Consider incorporating further references about clinical features of rickettsial diseases (see Biggs HM et al., 2016. doi: 10.15585/mmwr.rr6502a1; Adem V. 2019. Doi:10.1053/j.semdp.2019.04.005)
- Line 54. See Biggs HM et al., 2016. doi: 10.15585/mmwr.rr6502a1
- Line 59. The references (Begué, 2023; Li, 2021) you cited case reports (three cases as a whole) and because of that, little contribution to your point comes from such reports. The theoretical framework of your paper should be strengthened, I recommend seeing: Mosites et al. 2013. doi:10.4269/ajtmh.2012.12-0126; Bestul et al., 2022 doi: 10.4269/ajtmh.21-1017; Alvarez et al., 2018. Doi: 10.1093/trstmh/try030; Zientek et al., 2014. Doi:10.1016/j.jpeds.2013.10.008.
- Line 59. The author says, “There is currently no effective treatment available for this disease”, this statement is very inaccurate and must be corrected (See: Donovan et al., 2002. Doi: 10.1345/aph.1C089; Biggs HM et al., 2016. doi: 10.15585/mmwr.rr6502a1)
- Line 69. Ballman, 2023 does not refer to the China’s situation. Change the reference
- Line 74. State the CDC (acronym) meaning. Specify that Erlichiosis is the TBRD on which the selected areas were based, according to the reference you used.

Experimental design

Methods
Substantive changes need to be introduced in the study design. Some specific suggestions:

- Line 98. The study from Meher et al. (2022) is about food-borne diseases, not on TBRD. This must be changed to a one addressing at least some of the several rickettsial diseases. As in my opinion, this is a substantive flaw the manuscript as an entire scientific manuscript losses internal validity.
It is necessary to include the study population’ characteristics (selection criteria)
- Line 139 What was the result from the research map?

Validity of the findings

Line 98. The study from Meher et al. (2022) is about food-borne diseases, not on TBRD. This must be changed to a one addressing at least some of the several rickettsial diseases. As in my opinion, this is a substantive flaw the manuscript as an entire scientific manuscript losses internal validity.

Additional comments

None

Reviewer 2 ·

Basic reporting

Abstract:
I would suggest re-writing lines 27 – 30 in the abstract as currently the sentence is too long spanning multiple lines. Breaking up the sentences would help increase the clarity on the results obtained from the study. Also, recommend putting the number of participants who actually completed the survey rather than the number of people it was administered to, in line 23.
In line 31, authors mentioned that the ‘medical personnel were located in the first to fourth quadrants’. What does this mean? How does it relate to the results of the study?
Introduction:
The reference in line 44, is it Banovic et al 2023?
Methods:
Lines 93-96 not very clear as to what were the actual study sites from where participants were enrolled in the study. Please consider re-writing the sentence. Not sure about the need for a reference hospital.
Where is the reference for the sentence in line 99-100, “The sample size in the study area was determined according to the formula of the other author”.
The Cronback’s alpha reported in line 136, for which section of the questionnaire is that?
What is the protocol number for this study? Why do the authors mention using ArcGIS to make a research map in line 139 when no such map has been provided?
Results:
In line 146, authors have stated that women account for 59.12% of the respondents, however in table 1, female gender was reported for only 71 participants. Information is missing for 1,086 participants. In line 152, what kind of training did the participants receive? Was it general training on tick-borne diseases or training specifically given to participants for this study? In line 175, authors mention “people with a subtropical height and above”. What does this mean? This has been indicated in table 1 under the characteristic ‘Professional title’ as subtropical high and above which is unclear. Formatting and alignment of the tables provided are not consistent, making them hard to read. Authors could have provided some figures to show the participant scores for the Knowledge, Attitude and Practice sections, especially for the four-quadrant analysis that they have run.
Discussion:
Lines 211-215 do not make sense, since the referenced papers are about ticks, but authors talk about usage of mosquito repellants. The authors could have elaborated more on reasons behind the results they obtained in comparison to results from the literature.

Experimental design

The authors used and adapted the survey questionnaire from Meher et al 2022 which assessed knowledge and practices of street food vendors regarding food safety and attitudes and safe food handling practices. Whereas the target group for this study is medical staff and knowledge on tick-borne rickettsial diseases. The authors should have used survey instruments which were among the same demographic (that of medical practitioners or healthcare professionals) and were targeted to similar topics (that of tickborne or other vector-borne diseases) as demonstrated in Howard et al 2022 (https://doi.org/10.1177/10598405221099484), Carson et al 2022 (10.1016/j.onehlt.2022.100424), Mattoon et al 2021 (https://bmcinfectdis.biomedcentral.com/articles/10.1186/s12879-021-06622-6 ) etc. There should be valid reasoning provided as to how the authors used the survey from Meher et al 2022 and adapted it to conduct this study. The supplemental document attached named ‘questionnaire’ is not provided in English, making it difficult to see the survey instrument used in the study. It would have helped if the authors mentioned the inclusion criteria for participants used in the study, for example what was the minimum age? In table 1 the age categories are less than 30 years old, 30-40yrs and more than 40yrs.

Validity of the findings

The numbers provided in Table 1 do not add up to 1,206 (which is stated by the authors to be the total number of respondents who took the survey) for certain parameters specifically gender, age group, level of education, and professional title. Table 1 is really unreadable and hard to understand. When the authors were not able to get responses from all the respondents, they should mention that in the table. It is unclear if information is missing or if typos were made in creating some of these tables.

Additional comments

Being consistent with the use of hyphens, in some instances authors have used ‘tick borne’ in some cases ‘tick-borne,’ use one format and please use that format consistently throughout the manuscript. Also being consistent with the word ‘KAP,’ in the manuscript title it is KAPs whereas elsewhere in the manuscript it is written as KAP. The overall language used in the paper can be improved to increase clarity and readability of the manuscript. Please use consistent font and font sizes for the in-text references. There are several grammatical errors throughout the manuscript which need to be addressed. The paper is not clearly written and lacks essential information in the methods, results, and discussion sections.

---

## Round 0.2 · Minor Revisions

Please review the statistical methods and provide a more detailed interpretation of the findings to improve the results.

Reviewer 1 ·

Basic reporting

No comment

Experimental design

No comment

Validity of the findings

Section 4.4 The regression coefficients shown in table 4 should be interpreted to better understanding about the effect’s size of each predictor on KAP. It is understandable what of them were statistically significant but providing a brief interpretation in this section can improve the sense of your analysis.

“The results showed that awareness of TBRD was poor among medical staff, and interestingly, the average scores for attitudes and behaviors were greater than those for knowledge.” This statement is debatable, KAP constitutes a triad of interactive and interdependent factors, which can effectively change human health behaviors. Your finding suggests no connection among such factors, seemingly affected as separate components by individual characteristics of study subjects. You have provided a nice comparison of your results with several studies, but I encourage to give a better explanation about the above statement, may be by interpreting the results of your regression analysis.

Additional comments

Line 10. Provide correct reference

Provide reference for this statement: Here, an attitude was defined as “a complex mental state involving beliefs, feelings, values, and dispositions to act in certain ways”.

Provide explanation why the attitudes component was not incorporated into the analysis. “Multiple linear regression was carried out with the medical staff members’ knowledge of, and practices related to TBRD.”

Provide a brief explanation why you are using “attitudes” and “behavioral” as exchangeable terms across the manuscript. For theoretical consistency I recommend using KAP.

Annotated reviews are not available for download in order to protect the identity of reviewers who chose to remain anonymous.

---

## Round 0.3 · Minor Revisions

Here are some suggested edits in the attached file to improve the manuscript quality. Please review and submit a revised version at your convenience.

---

## Round 0.4 · accepted · Accept

Your manuscript is now accepted!